# The Benefits of Fibrinolysis Combined with Venous Systemic Oxygen Persufflation (VSOP) in a Rat Model of Donation after Circulatory Death and Orthotopic Liver Transplantation

**DOI:** 10.3390/ijms23095272

**Published:** 2022-05-09

**Authors:** Nadja Kröger, Zoltan Czigany, Jipin Jiang, Mamdouh Afify, Pascal Paschenda, Kazuyuki Nagai, Shintaro Yagi, René H. Tolba

**Affiliations:** 1Institute for Laboratory Animal Science and Experimental Surgery, Faculty of Medicine, RWTH-Aachen University, Pauwelsstraße 30, 52074 Aachen, Germany; nadja.kroeger@rwth-aachen.de (N.K.); zczigany@ukaachen.de (Z.C.); jpjiang@tjh.tjmu.edu.cn (J.J.); mafify@ukaachen.de (M.A.); ppaschenda@ukaachen.de (P.P.); kaznagai@kuhp.kyoto-u.ac.jp (K.N.); shintaro@kuhp.kyoto-u.ac.jp (S.Y.); 2Department of Plastic, Reconstructive and Aesthetic Surgery, Faculty of Medicine and University Hospital Cologne, University of Cologne, Kerpener Str. 62, 50937 Köln, Germany; 3Department of Surgery and Transplantation, Faculty of Medicine, University Hospital RWTH-Aachen, 52074 Aachen, Germany; 4Department of Surgery, Campus Charité Mitte/Campus Virchow-Klinikum, Charité-Universitätsmedizin, 13353 Berlin, Germany; 5Department of Pathology, Faculty of Veterinary Medicine, Cairo University, Giza 12211, Egypt

**Keywords:** orthotopic liver transplantation, organ preservation, ischemia-reperfusion injury, VSOP, oxygen, cold storage, DCD, donors after circulatory death, warm ischemia, fibrinolysis, streptokinase, microsurgery

## Abstract

Organ shortage has led to the increasing utilization of livers retrieved from donors after circulatory death (DCD). These pre-damaged organs are susceptible to further warm ischemia and exhibit minimal tolerance for cold storage. The aim was thus to examine the effects of fibrinolysis combined with Venous Systemic Oxygen Persufflation (VSOP) on the preservation of DCD livers in vivo. Livers of male Lewis rats were explanted after 45 min of warm ischemia, cold-stored for 18 h, and transplanted into a recipient animal. Livers were left untreated or underwent either VSOP or fibrinolysis via Streptokinase (SK) or received combined SK and VSOP. Combined treatment exhibited improved microvascular flow at 168 h (*p* = 0.0009) and elevated microperfusion velocity at 24 h post-transplantation (*p* = 0.0007). Combination treatment demonstrated increased portal venous flow (PVF) at 3 and 24 h post-transplantation (*p* = 0.0004, *p* < 0.0001), although SK and VSOP analogously achieved increases at 24 h (*p* = 0.0036, *p* = 0.0051). Enzyme release was decreased for combination treatment (*p* = 0.0002, *p* = 0.0223) and lactate dehydrogenase (LDH) measurements were lower at 24 h post-transplantation (*p* = 0.0287). Further supporting findings have been obtained in terms of serum cytokine levels and in the alterations of endothelial injury markers. The combination treatment of SK + VSOP might provide improved organ integrity and viability and may therefore warrant further investigation as a potential therapeutic approach in the clinical setting of DCD.

## 1. Introduction

Due to continuous improvements in patient selection, surgical techniques, perioperative care and the development of modern immunosuppressants liver transplantation (LT) became the treatment of choice for patients suffering from end-stage liver disease or acute liver failure [1,2,3]. As clinicians are faced with worldwide organ shortages due to insufficient numbers of organ donors, this challenge has led to the increasing utilization of marginal or extended criteria allografts [4,5,6,7]. Organs from donors after circulatory death (DCD) are exposed to warm ischemia and suffer a concomitant injury, which is linked to primary graft dys- or non-function and a high incidence of biliary complications after transplantation, which ultimately causes poor transplantation outcomes [8]. Molecular mechanisms associated with warm ischemia and therefore involved in graft damage include thrombosis, apoptosis, cytotoxic repercussion, the release of proinflammatory mediators, and Kupffer cell activation [9]. Multiple attempts have been made to thoroughly understand the process of ischemic injury and improve graft integrity after an ischemia-reperfusion injury [2,5,10,11,12,13,14]. The beneficial effect of graft reconditioning via Venous Systemic Oxygen Persufflation (VSOP) was already demonstrated in a porcine orthotopic LT model by Minor et al., as VSOP-treated grafts were able to show a sustained liver function and improved survival during an entire week after transplantation, whereas no survival could be achieved for recipients of non-treated grafts [15]. Subsequently, the principle of fibrinolysis prior to cold storage was shown to significantly improve tissue integrity and thus mitigate graft injury, which was validated, amongst other settings, in a DCD rat LT model by Yamauchi et al. [16]. The synergistic value of both treatments, namely fibrinolysis and hypothermic aerobic organ preservation via VSOP was finally shown by Tolba et al. in a DCD rat in vitro reperfusion model. Reduced portal venous pressure (PVP), less enzyme release, increased bile production, and less hepatocellular apoptosis upon reperfusion was demonstrated for liver grafts preserved with fibrinolysis and VSOP, suggesting improved viability after the aforementioned combined treatment [17]. Up until today, the effects of coupled VSOP and fibrinolysis on graft function and viability after LT are yet to be explored, therefore the goal of this study was to examine graft integrity and surgical outcome in a clinically relevant and complex microsurgical model of orthotopic LT following DCD in rats.

## 2. Results

### 2.1. Evaluation of Microcirculatory Perfusion

Figure 1A shows the microcirculatory flow measured using the O2C (O2C-Oxygen to see device and LF1 surface probe, LEA Medizintechnik GmbH, Giessen, Germany) technique. The control group exhibited the lowest microcirculatory flow at each timepoint. The combination treatment of Streptokinase (SK) and VSOP achieved the highest microcirculatory flow measurements at 24 h and 168 h post-transplantation (SK + VSOP 110.85 au ± 26.13, *p* < 0.0001; 101.06 au ± 18.53, *p* = 0.0009), whereas at 3 h post-transplantation, the sole treatment via VSOP displayed the highest values of microcirculatory flow (*p* = 0.0021). The treatment with SK only exhibited increased microcirculatory flow in comparison to the control group, but performed worse compared to the other treatments, except at 168 h post-transplantation, where higher absolute microcirculatory flow values were achieved in comparison to the treatment with VSOP only. Nevertheless, significant differences between the control group and the groups treated via SK solely could only be demonstrated at 24 h post-transplantation (*p* = 0.0025). The peak of microcirculatory flow was measured at 24 h post-transplantation for rats treated with combined SK and VSOP (*p* < 0.0001) and rats treated with VSOP only (*p* < 0.0001), indicating improved graft microcirculation after reperfusion for named treatments. Figure 1B displays the microperfusion velocity measured via the O2C measurement technique. The control group exhibited the lowest microcirculatory velocity at 3 h and 24 h post-transplantation, respectively, whereas at 168 h post-transplantation, no significant differences in microperfusion velocity could be shown for either of the groups. Rats that had received livers subjected to fibrinolysis only exhibited higher microperfusion velocity in comparison to the control group solemnly at 24 h post-transplantation (*p* = 0.0212). Significant differences in microperfusion velocity in comparison to the control groups could be demonstrated for rats that had received livers treated with VSOP only at 3 h and 24 h post-transplantation, respectively (*p* = 0.0311, *p* = 0.0041). The overall highest microperfusion velocity was measured at 24 h post-transplantation for recipients which had received livers treated with combined SK and VSOP (SK + VSOP 16.60 au ± 2.11, *p* = 0.0007), hereby indicating improved microcirculation after transplantation, particularly in comparison to the other treatment groups. Figure 1C demonstrates tissue oxygenation. The highest absolute values for tissue oxygenation were measured at 168 h post-transplantation for rats which had received livers treated with a combination of SK and VSOP, although no overall significant differences could be demonstrated for either of the groups at any given time point. Figure 1D shows the amount of intravascular hemoglobin measured within the grafts using O2C. All groups at all time points exhibited similar values. No significant differences could be demonstrated for either of the groups at any given time point.

### 2.2. Evaluation of Portal Circulation

Figure 2A shows the portal venous flow (PVF) measured via a transit-time perivascular flowmeter. The control group exhibited the lowest portal flow at all time points. Combined SK and VSOP resulted in the highest absolute portal flow values at all post-transplantation timepoints, whereas significant differences could be demonstrated only at 3 h and 24 h post-transplantation (SK + VSOP 13.6 mL/min ± 4.92, *p* = 0.0004; 18.8 mL/min ± 1.16, *p* < 0.0001), nevertheless indicating improved reperfusion. PVF increased over time reaching threshold values at approximately 18 mL/min at 168 h post-transplantation for each of the groups without exhibiting any significant differences between the experimental groups. Significant differences in comparison to the control group could also be demonstrated for recipients who have received livers preserved via VSOP at 3 h and 24 h post-transplantation sacrifice (*p* = 0.0481, *p* = 0.0051) and for recipients of livers treated via fibrinolysis only at 24 h post-transplantation sacrifice (*p* = 0.0036). Figure 2B displays the PVP measured via direct puncture of the portal vein. The overall highest values for PVP were measured at 168 h post-transplantation for all groups. The highest absolute PVP value was measured at approximately 13 mmHg within the control group, although no overall significant differences could be demonstrated for any of the groups at any given time point. At one week, the SK and VSOP group showed a tendency towards a lower PVP compared to the control group, however, it did not reach the level of statistical significance.

### 2.3. Serum GOT/AST, GPT/ALT, and LDH Measurements

Figure 3A shows the serum aspartate aminotransferase (GOT/AST) release measured at sacrifice. AST values peaked at 24 h post-transplantation in most groups. Strikingly, the combination of VSOP and SK led to strongly decreased AST levels compared to the other experimental groups (SK + VSOP, 645.8 IU ± 267.29, *p* = 0.0002 vs. control, *p* < 0.0001 vs. SK, *p* < 0.0001 vs. VSOP) which demonstrates a mitigated hepatocellular injury. Overall AST release decreased to nearly zero at 168 h post-transplantation in all groups. Figure 3B shows the serum alanine-aminotransferase (GPT/ALT) release. ALT values peaked at 3 to 24 h post-transplantation, depending on the experimental group, and decreased over time until overall ALT release reached nearly zero at 168 h post-transplantation for all groups. Combination SK and VSOP treatment showed significantly reduced ALT levels in comparison to the control group at 3 h and 24 h post-transplantation (SK + VSOP 1837 IU ± 254.33, *p* = 0.0268; 691.6 IU ± 117.44, *p* = 0.0223), thus indicating decreased hepatocellular damage for recipients of liver grafts pre-treated via fibrinolysis and VSOP. Rats that had received livers treated via combined SK and VSOP also displayed significantly reduced ALT release at 3 h and 24 h post-transplantation in comparison to the groups treated via SK alone (*p* = 0.0096, *p* = 0.0347). At 3 h post-transplantation, the group which had obtained livers treated via VSOP only exhibited the most pronounced decrease in ALT release in comparison to the control group, even with respect to the group which had received livers treated via combined SK and VSOP (*p* = 0.0025). Figure 3C shows the lactate dehydrogenase (LDH) release. Overall LDH values were measured highest at 3 h post-transplantation and decreased over time until LDH release reached nearly zero at 168 h post-transplantation among all groups. Most importantly, animals that had received livers treated via combined SK and VSOP demonstrated significantly reduced LDH release in comparison to animals of the control group, this becoming especially evident at 24 h post-transplantation (SK + VSOP 2208.6 IU ± 1458.87, *p* = 0.0287), hereby indicating markedly decreased mitochondrial graft damage. At 3 h post-transplantation, animals that had received livers preserved via VSOP solely exhibited significantly lower LDH release in comparison to the animals of the control group (*p* = 0.0033), whereas no significant difference could be demonstrated for animals that had obtained organs treated with combined SK and VSOP.

### 2.4. Histopathological Analysis

Table 1 shows the semi-quantitative histological scoring of livers of each group at 3 h and 24 h post-transplantation. At 3 h post-transplantation, the control group demonstrated an average general score of 1.8. The SK, as well as the SK + VSOP groups, demonstrated an average general score of 2.2, respectively. The group treated by VSOP amounted to an average general score of 2.8. Thus, at 3 h post-transplantation, the histological analysis did not indicate any hepatoprotective effects of named treatments. On the contrary, higher scores of the treated groups, namely the group which had received VSOP treatment, indicated a certain degree of parenchymal damage. At 24 h post-transplantation, the VSOP and SK + VSOP combination groups demonstrated markedly less damaged livers, reflected by average general histological scorings of 2 and 2.2, respectively, whereas the control group and the group treated via SK expressed hepatic damage, reflected by average general scores of 3.2 and 4, respectively. Generally speaking, the overall amount of completely unaffected tissue of full integrity was higher predominately among the groups which had received combination VSOP and SK treatment. Representative histological slides of the evaluated liver tissue of the different groups at different time points, as well as a graphical depiction of the general histological scores of the groups, are provided in Figure 4 (3 h/24 h: Control 1.8 ± 0.84, 3.2 ± 1.44; VSOP 2.8 ± 0.84, 3.8 ± 0.84; SK 2.2 ± 0.45, 4 ± 0.82; SK + VSOP 2.2 ± 0.84, 2.2 ± 0.45).

### 2.5. Serum TNF-α, IL-6, sICAM-1, HA and Immunohistochemical Analysis of RECA-1 Staining

Figure 5A shows tumor necrosis factor-alpha (TNF-α) release. TNF-α values were low or merely detectable at 3 h post-transplantation within all groups. At 24 h post-transplantation, the SK group exhibited the second highest TNF-α release following the control group. Nevertheless, no significant differences could be demonstrated for either of the groups. Figure 5B shows interleukin-6 (IL-6) release. IL-6 release was highest within the control and VSOP groups at 3 h post-transplantation, whereas at 24 h post-transplantation, IL-6 release was overall lower among all groups, except for the SK group. Analogously to TNF-α release, no significant differences could be demonstrated for either of the groups in terms of IL-6 release. Figure 5C shows soluble intercellular adhesion molecule-1 (sICAM-1) release. The highest sICAM-1 values were detected for the SK group at 3 h post-transplantation, whereas the least release was detected for the SK and VSOP combination group, resulting in a significant difference (*p* = 0.0288) between both groups. At 24 h post-transplantation, sICAM-1 release was similar for all of the groups. Figure 5D shows hyaluronic acid (HA) release. The highest HA release was measured within the control group at both, 3 h and 24 h post-transplantation. HA release slightly increased within the SK group at 24 h post-transplantation. Overall, no significant differences could be demonstrated for either of the groups. In sum, cytokine release seemed elevated especially within the control and SK groups. In contrast, the VSOP solely, as well as the SK and VSOP combination groups exhibited an overall diminished cytokine release, thus indicating a reduced proinflammatory response, although significant differences were not detected throughout the experiment. Figure 5E shows a graphical depiction of the histological scoring after immunohistochemical rat endothelial cell antibody-1 (RECA-1) staining of the liver samples. No significant difference in scoring was observed for either of the groups. Observed histological changes were within the moderate range at 3 h post-transplantation for all groups, whereas at 24 h post-transplantation, overall changes were less severe within all of the groups. The SK and VSOP combination group exhibited the overall lowest scoring at 24 h post-transplantation, thus suggesting the least endothelial damage.

## 3. Discussion

In this study, the effect of fibrinolysis combined with VSOP on the preservation of DCD livers was examined in the setting of liver transplantation in an isogenic rat model. Combining VSOP and fibrinolysis yielded beneficial effects, namely improved graft integrity with respect to microvascular perfusion and portal venous flow, as well as enzyme release. Histologically assessed graft damage was reduced in comparison to the other groups, although no improvements were achieved with respect to tissue oxygenation and portal venous pressure. Furthermore, this study highlights the timely shift of the treatment effect, as most outcomes started to become evident at 24 h post-transplantation and were mostly not measurable immediately after transplantation. The growing gap between the availability of donor allografts and the increasing number of patients registered for organ transplantation calls for new solutions to expand the available donor pool [18]. Due to improvements in organ preservation and retrieval techniques, DCD organ viability and transplantation outcomes for kidneys and livers have been continuously improved, which has led to the increasing use of DCD allografts in the battle against severe global organ shortage [19]. However, up until today, graft and patient survival for DCD LT remains inferior in comparison to donation after brain death (DBD) LT, thus emphasizing the continuous need for enhanced allograft treatment and organ preservation techniques to improve graft and patient outcomes [20]. The crucial factor for post-transplant viability lies in the period of warm ischemia, where sufficient perfusion before organ retrieval cannot be warranted. Further graft damage is then caused by the combination of subsequent cold ischemic storage and reperfusion injury following transplantation [21,22,23]. Livers retrieved from DCD are prone to severe perfusion deficits due to the formation of microthrombi in the setting of cardiac arrest, thus additionally hampering the appropriate equilibration of the graft microvasculature with the preservation solution [24,25]. The beneficial effects of temporary fibrinolysis, namely the improvement of graft microcirculation and oxygen supply have been demonstrated in multiple experimental animals, predominantly rat models, where a fibrinolytic preflush with SK resulted in a relevant and significant improvement of structural graft integrity as well as functional and metabolic graft recovery [25]. Analogously, the technique of VSOP has been shown to be an effective tool for resuscitating pre-damaged porcine livers after warm ischemic insult, thus allowing for successful transplantation after hypothermic storage [15]. In a rat transplantation model by Minor et al., preservation via VSOP lead to more homogenous aerobic conditions within the liver parenchyma and prevented the emergence of energy deficits during cold storage and therefore reduced parenchymal tissue injury upon reperfusion [26]. By combining VSOP and temporary fibrinolysis, Tolba et al. effectively preserved rat livers subjected to up to 90 min of warm ischemia, achieving results comparable to livers from heart-beating donors (HBD) with regards to enzyme release and bile production in an in vitro reperfusion model without LT [17]. Nevertheless, to the best of our knowledge, the in vivo effects of this combination therapy (SK + VSOP) on liver transplantation has never been demonstrated before. In this study, the previously reported results suggesting improved graft preservation via combined preservation strategies could be validated in a technically complex and elegant in vivo orthotopic LT model in rats: The combination of both strategies, VSOP and fibrinolysis, led to significant improvements in microvascular and portal venous flow, approximately doubling the amount of flow in comparison to the control groups and achieving the overall highest absolute flow values among all experimental groups. Although no significant differences in microperfusion velocity could be shown for either of the groups at 168 h post-transplantation, this is presumably due to a selection bias towards the animals who survived 168 h in the first place. Animals that had received grafts treated via combined VSOP and fibrinolysis exhibited markedly decreased AST and ALT enzyme release in comparison to the other animals, this becoming especially evident at the peak of reperfusion injury at 24 h post-transplantation. Furthermore, LDH release returned to near baseline at 24 h post-transplantation in the SK and VSOP group. These results altogether suggest improved blood circulation, enhanced graft integrity, and less hepatocellular injury. Histologically and immunohistochemically verified diminished parenchymatic and endothelial damage further supports the thesis of improved graft integrity after above mentioned combined treatment. The combination of VSOP and fibrinolysis thus represents a feasible and encouraging therapeutic approach for the transplantation of livers in the setting of DCD. Further research needs to be conducted in order to overcome the limitations of the animal model utilized and establish a transfer to clinically relevant liver storage and transplantation in human patients: Main drawbacks of the study, among others, include the differences of species and the respective differences in anatomy, molecular pathways, and operating procedure, the standardized cold-storage and warm ischemic times within the study, which does not reflect the clinical reality and the use of only male rats thus disrespecting gender diversity in transplant patients. Future research must therefore focus on further adaption to the current clinical setting and validation in other models and species to ultimately improve patient outcomes, as recipients of DCD livers still face poorer outcomes in comparison to recipients of DBD livers.

## 4. Materials and Methods

### 4.1. Animals

All experiments were performed in adherence to institutional guidelines, the EU Directive 2010/63, and the German federal law on the protection of animals. The respective authorities approved the ethical proposal of the study (LANUV Recklinghausen ID: 84-02.04.2012-A017). All animals received humane care conforming to the principles of the “Guide for the Care and Use of Laboratory Animals” (8th Edition, National Institutes of Health (NIH) Publication, 2011, USA). The study was designed, executed, and reported in line with the “Animal Research: Reporting of In Vivo Experiments” (ARRIVE) guidelines [27,28]. Male Lewis rats (Strain name: LEW/OrlRj, Janvier Labs, Le Genest Saint Isle, France; *n* (total) = 136 (68 recipient and 68 donor animals, body weight range: 175–200 g) were used in this study. As defined by the “Federation for Laboratory Animal Science Associations” (FELASA, www.felasa.eu, accessed on 4 August 2021), all animals were housed under specific pathogen-free (SPF) conditions. The barrier environment was temperature- and humidity-controlled and subjected to a 12 h light and dark cycle. Water and standard pellet food supply for laboratory rats (V1534-000 diet, Sniff GmbH, Soest, Germany) was provided ad libitum at all times.

### 4.2. Experimental Groups and Design

Liver transplantation was performed on a total number of 68 animals which was based on an a priori sample size calculation. Before the liver grafts were orthotopically transplanted into the recipient animal, they underwent 45 min of warm ischemia in situ to mimic a DCD situation and a subsequent cold storage for 18 h in 4 °C cold Histidine-tryptophan-ketoglutarate (HTK) solution supplemented with 20 mM of N-acetylcysteine (NAC, Hexal AG, Holzkirchen, Germany) in order to simulate a prolonged clinical transport and storage time of the graft. Animals were randomly allocated into four experimental groups. The control group (*n* = 20; 5 (3 h), 5 (24 h), 10 (168 h)) received livers subjected to cold storage in HTK without any additional treatment. Group 2 (*n* = 16; 5 (3 h), 5 (24 h), 6 (168 h)) received livers treated via VSOP during the time of cold storage: medical grade gaseous Oxygen was insufflated into the livers at a rate of 0.2 L/min with pressure being limited to a maximum of 18 mmHg via the suprahepatic vena cava (SHVC). Each liver lobe was punctured with two to three small pinpricks utilizing a customary acupuncture needle in order to let the gas emerge from the microvasculature. Group 3 (*n* = 16; 5 (3 h), 5 (24 h), 6 (168 h)) received livers that underwent a fibrinolytic preflush at room temperature (RT) via the portal vein containing 7500 international units (IU) of SK diluted in 20 mL of Sodium chloride (NaCl) immediately after warm ischemia and just before cold storage in HTK solution. Group 4 (*n* = 16; 5 (3 h), 5 (24 h), 6 (168 h)) received livers treated with both, VSOP and fibrinolytic preflush, as mentioned above. Recipient rats were sacrificed using lethal dose of anesthesia, organ retrieval, and subsequent exsanguination by blood drawing via direct puncture of the vena cava at 3, 24, and 168 h after reperfusion, respectively. All recipients survived 3 and 24 h after transplantation. In the control group, only 5 out of 10 recipients survived 168 h post-transplantation. In groups 2, 3, and 4, 5 out of 6 recipients survived 168 h. Based on our experience, regeneration of the transplanted liver grafts is usually almost complete after 168 h in all surviving animals independently of the treatment groups. Therefore, this subgroup was only used for the evaluation of recipient survival and further analyses were omitted in this particular subgroup.

After surgery, all animals were inspected at least every 12 h by an experienced veterinary technician blinded for the experimental design. Their clinical condition was concomitantly evaluated via a humane-endpoints score sheet designed specifically for liver transplantation research based on the previous works of Morton and Griffiths and Kanzler et al. [29,30]. Animals were euthanized when humane endpoints were reached. An overview of the study design is provided within the study design flowchart in Figure 6.

### 4.3. Surgical Techniques

All experiments were performed at the same time of the day in order to circumvent the possible influence of the circadian rhythm. Rats were allowed an acclimatization period of one week in our facility. All procedures were carried out according to the previously published recommendations on surgical procedures for rat models of LT by Nagai and Czigany et al. [28,31]. For a more detailed technical description, we, therefore, refer to the cited video manual and previous publications of our group. Inhalation anesthesia was applied utilizing 4 Vol% Isoflurane (Forane, Abbott GmbH, Wiesbaden, Germany) in 100% Oxygen at a flow rate of 4 L/min for induction and 1.5 Vol% Isoflurane in 100% Oxygen at a flow rate of 2 L/min for maintenance. A total of 0.1 mg/kg Buprenorphine (Temgesic, EssexPharma, Haar, Germany) was injected subcutaneously for analgesia. Rats were placed on a heating pad. All surgical procedures were carried out by the same surgeon. All surgical steps were performed under a surgical microscope. All intravenous injections were administered via the penile vein unless otherwise mentioned.

#### 4.3.1. Donor Procedure

Donor rats underwent midline laparotomy with bilateral extensions. The liver was freed from ligamentous attachments. The left phrenic vein was then ligated and cleaved. The paraoesophageal vessels were coagulated via bipolar forceps and subsequently dissected. After isolation of the infrahepatic vena cava (IHVC) from the retroperitoneal tissue, the right adrenal vein was ligated. According to the anatomical situation, either a 22- or 24-gauge catheter stent was proximally inserted into the bile duct via a small incision after ligation of the duct at the level of division of the gastroduodenal artery. For a detailed description of catheter stent preparation and insertion we again refer to the manuscript published by Nagai et al. [31]. The portal vein was then liberated from the pyloric and splenic veins. After ligating and cleaving the gastroduodenal artery, the common hepatic artery (CHA) was isolated from the pancreatic head to its root. The dorsal hepatic ligamentous attachments were then dissected. After approximately 3 min, the surgical field was re-exposed, cardiac arrest was induced via phrenotomy and the CHA was ligated proximal to its root. The IHVC and the portal vein were then both clamped. An 18-gauge catheter was then carefully inserted into the portal vein via a small incision and after 45 min of warm ischemia in situ, the liver was at first flushed with 20 mL of NaCl and subsequently perfused with 60 mL of 4 °C cold HTK solution supplemented with 20 mM of NAC at a hydrostatic pressure of 20 cm H_2_O in order to achieve an equilibration of the membrane potential. The diaphragm was then immediately cut, the intrathoracic vena cava transected and the anterior wall of the IHVC was surgically opened in order to allow the perfusion solution to be rinsed out of the liver. The IHVC was then clamped and the organ was then excised by dissection of the IHVC, the portal vein, the diaphragm, the remaining dorsal ligamental attachments, the right adrenal vein, and the CHA. For a detailed description of the dissection margins of the respective vessels, we refer to the manual previously published by Nagai et al. [31]. The retrieved graft was placed into 4 °C cold HTK solution supplemented with 20 mM of NAC and stored in a metal cup mounted in a plastic box filled with crushed ice.

#### 4.3.2. Ex Vivo Graft Preparation after Retrieval

Ex vivo graft preparations were all performed which included the placement of a 14-gauge cuff to the portal vein, the insertion of a 24-gauge catheter stent into the CHA, and the placement of a 12-gauge cuff to the IHVC. For a more detailed and depicted description of cuff and catheter stent preparation and placement, we refer to our manual published by Nagai et al. [31]. The liver was subsequently flushed with 5 mL of 4 °C cold HTK solution via the arterial catheter stent. In order to adequately prepare the SHVC for latter transplantation, two 7-0 polypropylene sutures were placed as stay sutures for later anastomosis at both corners of the vein, respectively. For two animals, placement of a 12-gauge cuff to the IHVC was not feasible. In these cases, 14-gauge cuffs were utilized for IHVC reconstruction. Organs were then stored in a cold-water bath containing HTK solution at a temperature of 4 °C for 18 h in order to simulate transportation time of the graft.

#### 4.3.3. Recipient Procedure

Rats underwent midline laparotomy. The liver was then freed from ligamentous attachments. The left phrenic vein was then ligated and cleaved. The IHVC was then carefully isolated from the retroperitoneal tissue. Next, the right adrenal vein was ligated and cut. The dorsal hepatic ligamentous attachments and the bile duct were dissected. Subsequently, the gastroduodenal artery and proper hepatic artery were ligated and dissected at 3 mm distance from their origination of the CHA. Remaining dorsal ligamentous attachments were cleaved. After intravenous injection of 2 mL lactated Ringer solution, the IHVC was clamped. The portal vein was then also clamped, as well as the SHVC together with the corresponding diaphragm. During anhepatic, portal, and arterial cross-clamping time, Isoflurane anesthesia was reduced to 0.4 Vol%. The native liver was subsequently excised and the graft was then placed orthotopically into the surgical site. Transplantation was commenced by anastomosis of the SHVC. Next, the portal vein was anastomosed via cuff technique. Clamping of the portal vein and SHVC was then released, leading to liver reperfusion. Isoflurane inhalation anesthesia was then again increased to 0.8 Vol%. The procedure was continued by reconstructing the CHA by catheter stent technique. The previously placed clamp was then released. The recipient IHVC was then reconstructed by cuff technique. After declamping, Isoflurane inhalation anesthesia was again increased to 1.0 Vol% and 0.5 mL of 8.4% NaCl-Bicarbonate solution together with 1 mL of lactated Ringer solution were administered intravenously. Afterward, the reconstruction of the bile duct by catheter stent technique was performed. Upon thorough completion of reconstruction, 1 mL of 5% Glucose solution was administered intravenously. Animals were supplied with a subcutaneous injection of Cefuroxime-NaCl (16 mg/kg) and Buprenorphine (0.1 mg/kg) diluted in a total of 1.5 mL of regular Saline solution immediately after surgery. Rats were then placed in a specialized intensive care unit cage with heated air (30–35 °C) and Oxygen supply for 1 h. Buprenorphine (0.1 mg/kg) was injected subcutaneously every 12 h for 3 d. Rats were then transferred to their normal cages and provided food and water ad libitum again. Animals were sacrificed after 3, 24, and 168 h after transplantation, respectively. For a more detailed description of suturing techniques and reconstruction techniques via cuff or catheter stent, we repeatedly refer to the manual published by our group [31]. Please note that in the present study the IHVC was reconstructed via cuff, rather than suture technique.

### 4.4. Anhepatic Time and Survival Estimate

Anhepatic time (Figure 7A) within the recipient procedure was registered for each surgical case in order to ensure comparability of the performed measurements and investigated results between the different groups. No significant differences in anhepatic time between the groups could be demonstrated (mean anhepatic time in all groups: 17.75 min). A Kaplan-Meier estimate of recipient survival for better understanding and transparency in drop-out and survival for each group is provided in Figure 7B.

### 4.5. Evaluation of Microcirculatory Perfusion

Hepatic microcirculatory perfusion was evaluated at sacrifice, just before the measurement of portal circulation and the collection of blood and tissue samples. Relative microcirculatory blood flow, blood flow velocity, tissue oxygen saturation, and relative hemoglobin amount were evaluated with O2C measurement technique via a corresponding surface probe in order to investigate graft perfusion. Measurements were performed at four standardized reference points of the liver surface and the mean value was calculated in order to characterize microcirculation, as previously described by Czigany et al. [10]. The captured signals were then transferred to an integrated computer, where software enabled the real-time display of the data with an accompanying visualization of the blood flow pattern (Integrated factory software, LEA Medizintechnik GmbH, Giessen, Germany).

### 4.6. Evaluation of Portal Circulation

PVF and PVP were evaluated at the time of sacrifice immediately before collecting blood and tissue samples, as previously proposed by Yagi et al. [32]. PVF was therefore measured with a transit-time perivascular flowmeter (T403, Transonic Systems Inc., Ithaca (NY), USA) utilizing a transonic flow probe (MA2PSB, Transonic Systems Inc., Ithaca (NY), USA) and PVP was measured via direct puncture of the anterior wall of the portal vein with a 27-gauge needle (BD Microlance 3, Becton Dickinson GmbH, Heidelberg, Germany) and subsequent recording with a corresponding monitoring device (Sirecust 404, Siemens, Erlangen, Germany). PVF and PVP were investigated to assess gross vascular perfusion.

### 4.7. Serum GOT/AST, GPT/ALT, and LDH Measurements

Blood samples were collected from the IHVC by direct puncture with a customary 20-gauge needle during sacrifice. After centrifugation at RT for 10 min at 2500 rpm, serum levels of AST and ALT as markers for hepatocellular injury were measured via an automated analyzer (Vitros 250, Johnson and Johnson, Neuss, Germany). Furthermore, serum levels of lactate dehydrogenase (LDH) as an index for mitochondrial graft damage and cell death were measured analogously.

### 4.8. Histopathological Analysis

Retrieved livers were macroscopically examined and immediately fixed in 10% neutral buffered formalin and consecutively embedded in paraffin. Microtome slides of 4 μm thickness were cut and subsequently stained via hematoxylin and eosin (HE) dye and then investigated in a blinded fashion by a senior pathologist. The histological examiner was blinded for the experimental set-up, respective treatments, and animal grouping. Livers of animals sacrificed at 3 h and 24 h post-transplantation were investigated. Ten randomly chosen and non-overlapping fields at a 400-fold magnification were chosen in order to be evaluated via light microscopy (Leica Digital Microscope 2500, Leica Microsystems GmbH, Wetzlar, Germany). A semi-quantitative scoring system based upon a formerly published scoring assessment by Streidl et al. in 2021 was employed for the analysis of hepatic injury [33]: Histological signs of injury, namely hemorrhage, congestion, inflammatory cell infiltration, necrosis, degenerative changes, biliary epithelial proliferation, and Kupffer cell activation were assessed and graded on a scale ranging from 1–5, whereas a score of 1 represented either no histological changes at all or overall negligible lesions, a score of 2 = rather mild lesions, a score of 3 = moderate lesions, a score of 4 = moderate to severe lesions and a score of 5 resembled severe lesions. A general score for each slide was then calculated and the general scores for each respective group were then averaged. The approximate amount of tissue not affected at all within each section was also assessed.

### 4.9. ELISA Measurements of TNF-α, IL-6, sICAM-1, HA, and Immunohistochemical Analysis of RECA-1 Staining

Commercially available rat enzyme-linked immunoassay (ELISA) kits (R and D Systems, Minneapolis, MN, USA) were utilized and carried out according to the manufacturer’s instructions: Serum samples were, first of all, stored at −80 °C and subsequently used for TNF-α, IL-6, sICAM-1, and HA assessment. In addition, liver samples at 3 h and 24 h post-transplantation were immunohistochemically stained with RECA-1 according to the manufacturer’s instructions and then forwarded to the respective pathologist for further examination. All slides were investigated in a blinded fashion by a senior pathologist. Six vessels were randomly chosen per slide and further investigated at a 400-fold magnification via upright light microscopy (Leica Digital Microscope 2500). A semi-quantitative scoring system was employed for the analysis of vascular integrity, whereas a score of 0 represented either no histological changes at all or overall negligible lesions, corresponding to thin and intact endothelium without any disruption and normal lining, a score of 1 represented rather mild lesions with slightly thicker endothelium and only few disruptions, a score of 2 represented moderate lesions and a score of 3 represented rather severe lesions. A score for each slide was then calculated and the scores for each respective group were then averaged.

### 4.10. Statistical Analysis

All results are expressed as mean values ± standard error of the mean (SEM) or respectively, standard deviations (SD), for each group. Two-way analysis of variance (ANOVA) and Tukey post hoc tests were performed to analyze changes in time-dependent parameters and differences between each group at each time point. Kaplan-Meier survival estimate and the log-rank test were utilized to analyze and depict recipient survival. Differences were considered significant when *p* < 0.05. Data analysis and plotting were performed using the GraphPad Prism 8 software package (GraphPad Prism 8 software package, GraphPad Software Inc., San Diego, CA, USA).

## Figures and Tables

**Figure 1 ijms-23-05272-f001:**
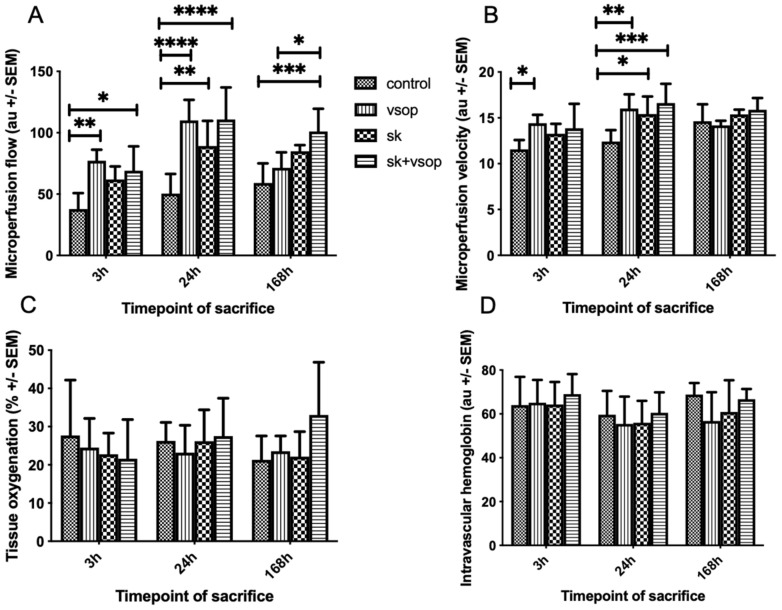
(**A**) Microvascular flow of the liver grafts at 3, 24 and 168 h post-transplantation in arbitrary units (au, see text). The control group exhibited the least microvascular flow. Significant differences could be demonstrated particularly for the groups treated either with Streptokinase (SK) and Venous Systemic Oxygen Persufflation (VSOP) or only VSOP at sacrifice 24 h post-transplantation (* = *p* < 0.05, ** = *p* < 0.01, *** = *p* < 0.001, **** = *p* < 0.0001). Standard error of the mean (SEM). (**B**) Microvascular velocity of the liver grafts at 3, 24 and 168 h post-transplantation in arbitrary units (au, see text). Significant differences could be demonstrated particularly for the groups treated either with Streptokinase (SK) and Venous Systemic Oxygen Persufflation (VSOP) or only VSOP at sacrifice 24 h post-transplantation (* = *p* < 0.05, ** = *p* < 0.01, *** = *p* < 0.001, **** = *p* < 0.0001). Standard error of the mean (SEM). (**C**) Tissue Oxygen saturation of the liver grafts at 3, 24 and 168 h post-transplantation in percent (%, see text). There was no significant difference (* = *p* < 0.05) in tissue Oxygen saturation between the four groups at any timepoint of sacrifice after transplantation. Standard error of the mean (SEM). (**D**) Relative intravascular amount of hemoglobin of the liver grafts at 3, 24 and 168 h post-transplantation in arbitrary units (au, see text). There was no significant difference (* = *p* < 0.05) in relative intravascular amount of hemoglobin between the four groups at any timepoint of sacrifice after transplantation. Standard error of the mean (SEM).

**Figure 2 ijms-23-05272-f002:**
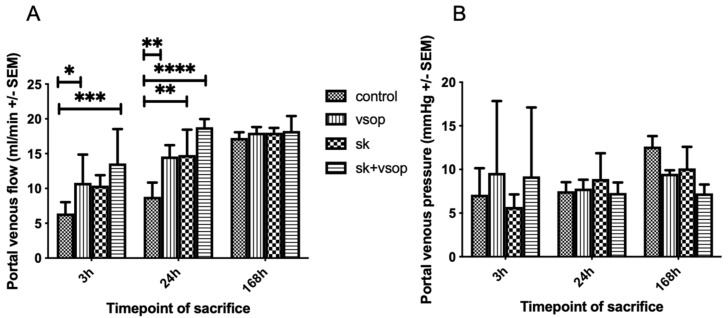
(**A**) Portal venous flow (PVF) of the liver grafts at 3, 24 and 168 h post-transplantation in milliliter/minute (mL/min, see text). The control group exhibited the least PVF. Significant differences could be demonstrated particularly for the groups treated with Streptokinase (SK) and Venous Systemic Oxygen Persufflation (VSOP) at sacrifice 3 h and 24 h post-transplantation, respectively (* = *p* < 0.05, ** = *p* < 0.01, *** = *p* < 0.001, **** = *p* < 0.0001). Standard error of the mean (SEM). (**B**) Portal venous pressure (PVP) of the liver grafts at 3, 24 and 168 h post-transplantation in millimeter of Mercury (mmHg, see text). There was no significant difference (* = *p* < 0.05) in PVP between the four groups at any timepoint of sacrifice after transplantation. Standard error of the mean (SEM).

**Figure 3 ijms-23-05272-f003:**
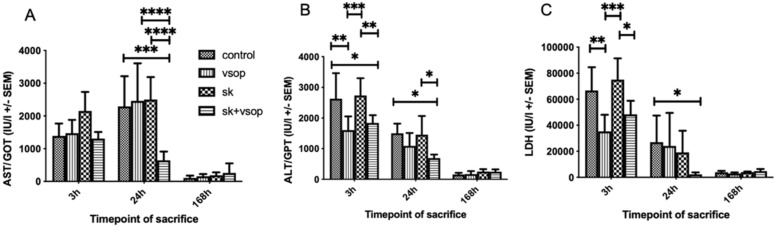
(**A**) Glutamic oxaloacetic transaminase (GOT/AST) release of the liver grafts at 3, 24 and 168 h post-transplantation in international units (IU, see text). AST release peaked at sacrifice 24 h post-transplantation. Significant differences could be demonstrated particularly for the group treated with Streptokinase (SK) and Venous Systemic Oxygen Persufflation (VSOP) at sacrifice 24 h post-transplantation, as this group exhibited unanticipated low AST release in comparison to the other groups at named timepoint. (* = *p* < 0.05, ** = *p* < 0.01, *** = *p* < 0.001, **** = *p* < 0.0001). Standard error of the mean (SEM). (**B**) Glutamic pyruvic transaminase (GPT/ALT) release of the liver grafts at 3, 24 and 168 h post-transplantation in international units (IU, see text). ALT release peaked at sacrifice 3 h post-transplantation. Significant differences could be demonstrated particularly for the groups treated either with Streptokinase (SK) and Venous Systemic Oxygen Persufflation (VSOP) or only VSOP at sacrifice 3 h post-transplantation, as these groups exhibited lower ALT release in comparison to the other groups at named timepoint. (* = *p* < 0.05, ** = *p* < 0.01, *** = *p* < 0.001, **** = *p* < 0.0001). Standard error of the mean (SEM). (**C**) Lactate dehydrogenase (LDH) release of the liver grafts at 3, 24 and 168 h post-transplantation in international units (IU, see text). LDH release peaked at sacrifice 3 h post-transplantation. Significant differences could be demonstrated particularly for the group treated with Venous Systemic Oxygen Persufflation (VSOP) at sacrifice 3 h post-transplantation, as this group exhibited lower LDH release in comparison to the other groups at named timepoint. Significant differences could also be demonstrated for the group treated with Streptokinase (SK) and VSOP at sacrifice 24 h post-transplantation, as this group exhibited unanticipated low LDH release in comparison to the other groups at named timepoint. (* = *p* < 0.05, ** = *p* < 0.01, *** = *p* < 0.001, **** = *p* < 0.0001). Standard error of the mean (SEM).

**Figure 4 ijms-23-05272-f004:**
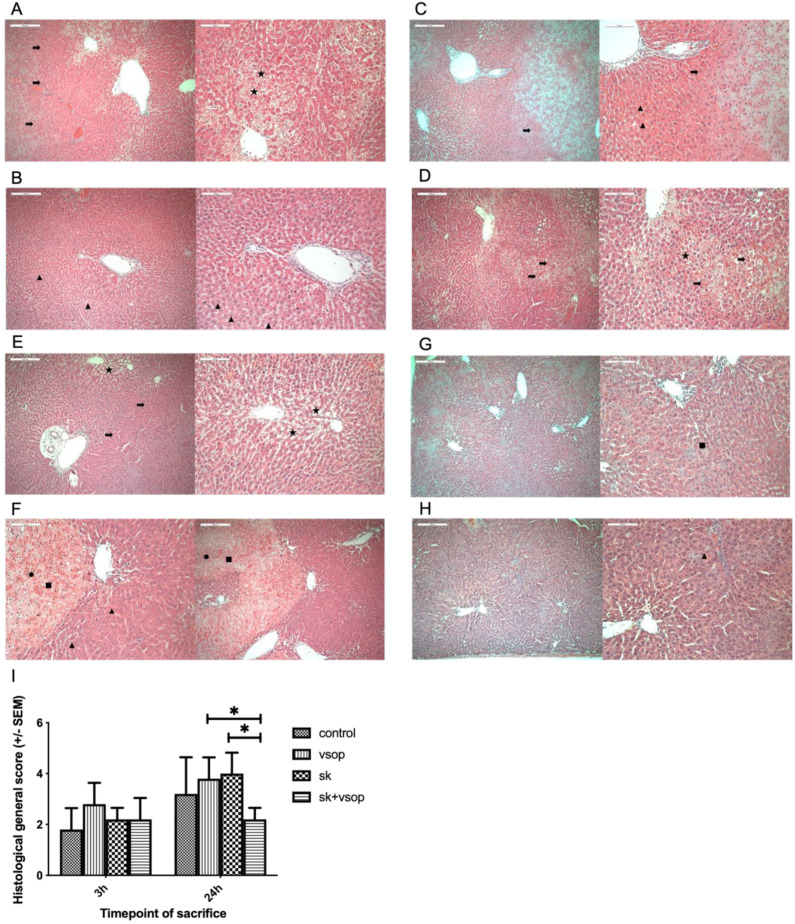
(**A**) Histopathologic example of the control group at 3 h post-transplantation. Congestion (arrow). Macrovesicular fatty vacuolization (star). (**B**) Representative histopathology of the Streptokinase (SK) group at 3 h post-transplantation. Micro vesicular fatty vacuolization (triangle). (**C**) Example of histopathology of the Venous Systemic Oxygen Persufflation (VSOP) group at 3 h post-transplantation. Congestion (arrow). Micro vesicular fatty vacuolization (triangle). (**D**) Exemplary histopathology of the SK and VSOP group at 3 h post-transplantation. Congestion (arrow). Macrovesicular fatty vacuolization (star). (**E**) Exemplary histopathology of the control group at 24 h post-transplantation. Congestion (arrow). Macrovesicular fatty vacuolization (star). (**F**) Exemplary histopathology of the SK group at 24 h post-transplantation. Micro vesicular fatty vacuolization (triangle). Confluent necrotic areas (square). Mononuclear cell infiltration (dot). (**G**) Exemplary histopathology of the VSOP group at 24 h post-transplantation. Confluent necrotic areas (square). Overall less parenchymatic damage. (**H**) Exemplary histopathology of the SK and VSOP group at 24 h post-transplantation. Micro vesicular fatty vacuolization (triangle). Overall less parenchymatic damage. (**I**) Histological general score of the liver grafts at 3 and 24 h post-transplantation. At 24 h post-transplantation, the SK and VSOP group exhibited an overall lower score than all of the other groups, thus indicating less parenchymatic damage, although no significant difference could be detected in comparison to the control group. (* = *p* < 0.05). Standard error of the mean (SEM).

**Figure 5 ijms-23-05272-f005:**
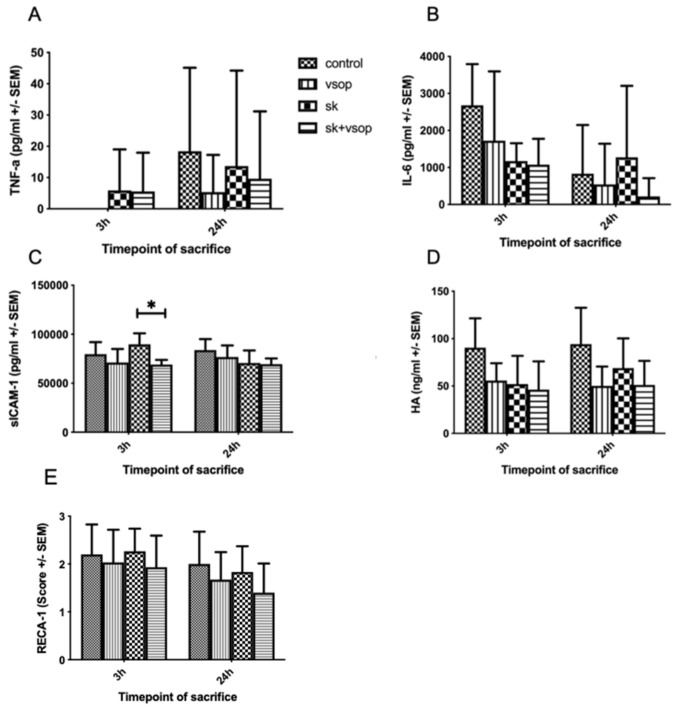
(**A**) Tumor necrosis factor alpha (TNF-α) release of the liver grafts at 3 and 24 h post-transplantation in picogram/milliliter (pg/mL, see text). There was no significant difference (* = *p* < 0.05) in between the four groups at any timepoint of sacrifice after transplantation. Standard error of the mean (SEM). (**B**) Interleukin-6 (IL-6) release of the liver grafts at 3 and 24 h post-transplantation in picogram/milliliter (pg/mL, see text). There was no significant difference (* = *p* < 0.05) in between the four groups at any timepoint of sacrifice after transplantation. Standard error of the mean (SEM). (**C**) Soluble intercellular adhesion molecule-1 (sICAM-1) release of the liver grafts at 3 and 24 h post-transplantation in picogram/milliliter (pg/mL, see text). At 3 h post-transplantation, the Streptokinase (SK) group exhibited an overall higher release of sICAM-1 than the other groups, thus resulting in a significant difference in comparison to the SK and Venous Systemic Oxygen Persufflation (VSOP) group. (* = *p* < 0.05). Standard error of the mean (SEM). (**D**) Hyaluronic acid (HA) release of the liver grafts at 3 and 24 h post-transplantation in nanogram/milliliter (ng/mL, see text). There was no significant difference (* = *p* < 0.05) in between the four groups at any timepoint of sacrifice after transplantation. Standard error of the mean (SEM). (**E**) Rat endothelial cell antibody-1 (RECA-1) immunohistochemical staining scoring of the liver grafts at 3 and 24 h post-transplantation. There was no significant difference (* = *p* < 0.05) in between the four groups at any timepoint of sacrifice after transplantation. Standard error of the mean (SEM).

**Figure 6 ijms-23-05272-f006:**
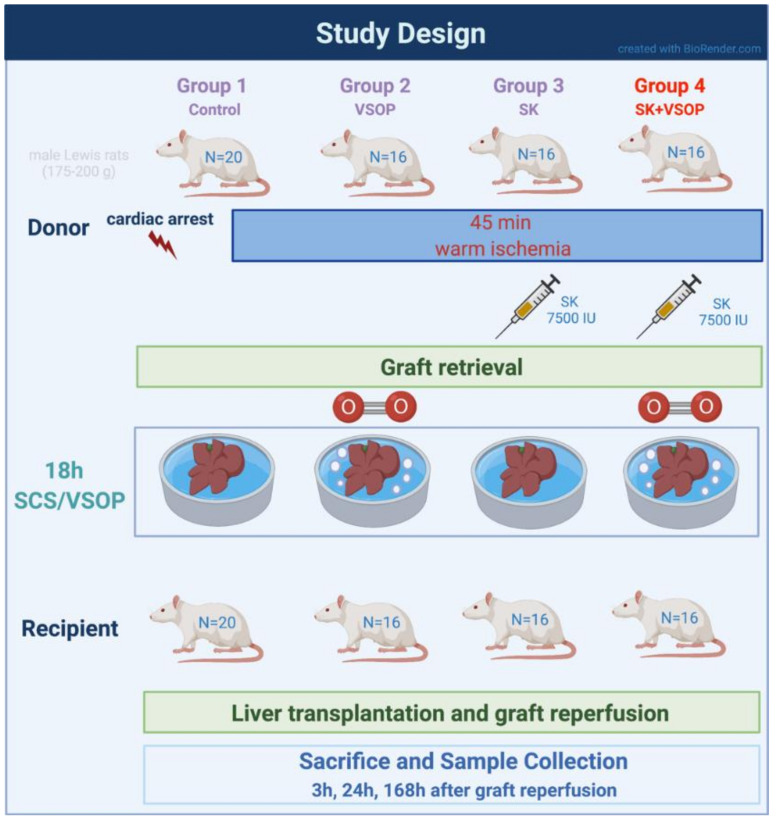
Flowchart of the study design. Created with BioRender.com.

**Figure 7 ijms-23-05272-f007:**
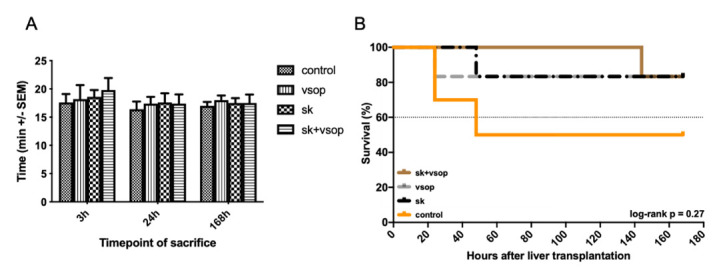
(**A**) Anhepatic time of the liver grafts as part of the recipient procedure in minutes (min, see text). There was no significant difference in anhepatic time between the four groups at any timepoint of sacrifice after transplantation. Mean +/- Standard error of the mean (SEM). (**B**) Kaplan-Meier estimate of recipient survival for each group under consideration of the three investigated timepoints at 3 h, 24 h and 168 h post-transplantation. Streptokinase (SK) and Venous Systemic Oxygen Persufflation (VSOP) combination group exhibited increased survival with the least drop-outs, whereas the control group displayed lower survival especially within the first 40 h post-transplantation. The log-rank test showed no significant differences (*p* = 0.27).

**Table 1 ijms-23-05272-t001:** Semi-quantitative histological scoring of livers of each group at 3 h (top half) and 24 h (bottom half) post-transplantation. Approx. areas without lesions = Approximate percentage of areas without significant lesions in the section. Scoring ranging from 1–5, whereas a score of 1 represents either no histological changes at all or overall negligible lesions, a score of 2 = rather mild lesions, a score of 3 = moderate lesions, a score of 4 = moderate to markedly severe lesions and a score of 5 = severe lesions. General scoring is provided on the far right. Averages of the general scores and respective standard deviations (SD) are provided beneath each group package, as indicated above.

	Group	Approx. Areas without Lesions	Hemorrhage	Congestion	Inflammatory Cell Infiltration			Necrosis	Degenerative Changes		Biliary Epithelial Proliferation	Sinu. Endothelial/ Kupffer Cell Activation	General Score
					Granulocytes	Macrophages	Mononuclear Cells (Other)		Fatty Degeneration	Granularity of Cytoplasm			
**3 h**	**Control**	50	1	4	1	1	1	1	1	2	1	1	2
		70	1	3	1	1	1	1	1	1	1	1	2
		95	1	2	1	1	1	1	1	1	1	1	1
		85	1	2	1	1	1	1	1	2	1	1	1
		60	1	4	1	1	1	2	3	3	1	1	3
	Mean/SD	72 +/− 18	1	3 +/− 1	1	1	1	1.2 +/− 0.44	1.4 +/− 0.89	1.8 +/− 0.84	1	1	**1.8 +/− 0.84**
**3 h**	**SK**	40	1	3	1	1	1	1	2	3	1	3	2
		60	1	3	1	1	1	1	2	4	1	1	3
		75	1	3	1	1	1	1	1	3	1	1	2
		60	1	2	1	1	1	1	1	2	1	1	2
		75	1	2	1	1	1	1	2	3	1	1	2
	Mean/SD	62 +/− 14	1	2.6 +/− 0.55	1	1	1	1	1.6 +/− 0.55	3 +/− 0.71	1	1.4 +/− 0.89	**2.2 +/− 0.45**
	**SK + VSOP**	40	2	4	2	1	1	2	2	2	1	1	3
**3 h**		70	1	3	1	1	1	2	2	2	1	1	3
		85	1	2	1	1	1	1	2	2	1	1	2
		80	1	3	1	1	1	1	3	2	1	1	2
		90	1	2	1	1	1	1	2	2	1	1	1
	Mean/SD	73 +/− 19	1.2 +/− 0.45	2.8 +/− 0.84	1.2 +/− 0.45	1	1	1.4 +/− 0.55	2.2 +/− 0.45	2	1	1	**2.2 +/− 0.84**
	**VSOP**	90	1	3	1	1	1	1	2	2	1	1	2
**3 h**		60	1	3	1	1	1	2	2	2	1	1	3
		60	2	4	3	1	1	4	2	3	1	1	4
		60	1	3	1	1	1	3	2	3	1	2	3
		85	1	2	1	1	1	1	2	2	1	1	2
	Mean/SD	71 +/− 15	1.2 +/− 0.45	3 +/− 0.71	1.4 +/− 0.89	1	1	2.2 +/− 1.30	2	2.4 +/− 0.55	1	1.2 +/− 0.45	**2.8 +/− 0.84**
	**Control**	95	1	2	1	1	1	1	2	2	1	2	1
**24 h**		60	1	4	3	2	1	3	3	3	2	3	4
		60	1	3	2	2	1	3	2	3	3	2	3
		55	2	4	5	3	2	4	3	3	3	2	5
		75	2	3	2	2	2	3	2	2	3	3	3
	Mean/SD	69 +/− 16	1.4 +/− 0.55	3.2 +/− 0.84	2.6 +/− 1.52	2 +/− 0.71	1.4 +/− 0.55	2.8 +/− 1.10	2.4 +/− 0.55	2.6 +/− 0.55	2.4 +/− 0.90	2.4 +/− 0.55	**3.2 +/− 1.44**
	**SK**	40	2	4	3	2	2	5	3	3	3	2	5
**24 h**		60	2	4	4	2	2	4	3	3	3	2	4
		85	1	3	3	2	2	3	2	3	3	2	3
		65	1	4	3	2	2	4	2	3	3	2	4
	Mean/SD	63 +/− 18.48	1.5 +/− 0.58	3.75 +/− 0.5	3.25 +/− 0.5	2	2	4 +/− 0.82	2.5 +/− 0.58	3	3	2	**4 +/− 0.82**
	**SK + VSOP**	90	1	2	1	1	1	2	2	2	2	1	2
**24 h**		90	1	2	2	1	1	2	2	2	2	2	2
		85	1	2	4	2	1	4	2	2	2	2	3
		90	1	2	2	2	1	2	2	2	2	2	2
		95	1	2	2	2	1	2	2	2	2	2	2
	Mean/SD	90 +/− 3.54	1	2	2.2 +/− 1.1	1.6 +/− 0.55	1	2.4 +/− 0.9	2	2	2	1.8 +/− 0.45	**2.2 +/− 0.45**
	**VSOP**	75	1	3	3	2	1	4	2	3	2	2	4
**24 h**		40	2	4	3	2	1	4	2	3	2	2	5
		60	2	4	3	2	1	4	3	3	2	2	4
		85	1	3	2	2	1	3	2	3	2	2	3
		90	2	3	2	2	1	3	2	3	2	2	3
	Mean/SD	70 +/− 20.31	1.6 +/− 0.55	3.4 +/− 0.55	2.6 +/− 0.55	2	1	3.6 +/− 0.55	2.2 +/− 0.45	3	2	2	**3.8 +/− 0.84**

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
