# Peer review of "The Benefits of Fibrinolysis Combined with Venous Systemic Oxygen Persufflation (VSOP) in a Rat Model of Donation after Circulatory Death and Orthotopic Liver Transplantation"

_ijms, 2022, doi:10.3390/ijms23095272_

Round 1
Reviewer 1 Report
The authors examined the synergistic value of two different treatment strategies, namely fibrinolysis and hypothermic aerobic organ preservation via VSOP, for the preservation of DCD livers in an in vivo orthotopic liver transplantation model in rats. The manuscript is well-written and presents the performed research concisely and comprehensively.
Minor text-editing is necessary: Some abbreviations (e.g. O2C, PVF) are used while the full term for which the abbreviation stands did not precede its first use in the text. In the legend of figure 4, some symbols are missing in the PDF version of the manuscript. With regards to terminology, organ "retrieval" or "procurement" is the preferred term, instead of "harvesting" (discussion section). Line 346, please use a verb instead of the gerund "indicating". The line drawing of figure 7B is difficult to discriminate, especially in the inserted legend.
The authors should comment on why livers of animals sacrificed at 168 h post-transplantation were not investigated.
Reviewer 2 Report
The work of Nadja Kröger et al. entitled ‘The benefits of fibrinolysis combined with Venous Systemic Oxygen Persufflation (VSOP) in a rat model of donation after 3 circulatory death and orthotopic liver transplantation’ over all sound and clearly point out the combined benefit. The manuscript is clearly structured and the used methods are state of the art. The obtained results are not overinterpret and presented clearly.
Overall, the manuscript did not need in depth revisions. There are only minor revisions necessary:
- In Figure 4 the symbols for the given histopathologic examples are missing - empty brackets.
- As there are high derivation in Figure 5 (most likely A, B and D) a dot blot instead of a column would be the more preferable way to present the results. Showing the single measurements would improve the results in term of interpretation of outliers, which may ‘hidden’ in a column.
- As biliary complications are important factor for transplantation outcome, a more detailed analysis of the hepatobiliary system would improve the work. The authors may investigate the conjugated and unconjugated bilirubin as a marker for functional impaired canalicular transport. Also, the histopathologic analyses of the apical canalicular transporter and the basolateral transporter may underline the effect of the combined fibrinolysis and VSOP administration.
Best regards
